# Recovery after human bone marrow mesenchymal stem cells (hBM-MSCs)-derived extracellular vesicles (EVs) treatment in post-MCAO rats requires repeated handling

Yolanda Gomez-Galvez[1,2,3], Malvika Gupta[4], Mandeep Kaur[4], Salvatore Fusco[5,6], Maria Vittoria Podda[5,6], Claudio Grassi[5,6], Amit K. Srivastava[4], Lorraine Iacovitti[1,2,3,7]*, Elena Blanco-Suarez[1,2,3,7]*

1 Department of Neuroscience, Sidney Kimmel Medical College, Thomas Jefferson University, Philadelphia, Pennsylvania, United States of America, 2 The Joseph and Marie Field Laboratory for Cerebrovascular Research, Sidney Kimmel Medical College, Thomas Jefferson University, Philadelphia, Pennsylvania, United States of America, 3 Vickie & Jack Farber Institute for Neuroscience, Sidney Kimmel Medical College, Thomas Jefferson University, Philadelphia, Pennsylvania, United States of America, 4 Division of Hematology, Department of Medicine, Cardeza Foundation for Hematologic Research, Sydney Kimmel Medical College, Thomas Jefferson University, Philadelphia, Pennsylvania, United States of America, 5 Department of Neuroscience, Università Cattolica del Sacro Cuore, Rome, Italy, 6 Fondazione Policlinico Universitario Agostino Gemelli IRCCS, Rome, Italy, 7 Department of Neurological Surgery, Sidney Kimmel Medical College, Thomas Jefferson University, Philadelphia, Pennsylvania, United States of America

* elena.blancosuarez@jefferson.edu (EB-S); lorraine.iacovitti@jefferson.edu (LI)

**Data Availability Statement:** All relevant data are within the manuscript and its Supporting information files.

## Abstract

Rehabilitation is the only current intervention that improves sensorimotor function in ischemic stroke patients, similar to task-specific intensive training in animal models of stroke. Bone marrow mesenchymal stem cells (BM-MSCs)-derived extracellular vesicles (EVs) are promising in restoring brain damage and function in stroke models. Additionally, the non-invasive intranasal route allows EVs to reach the brain and target specific ischemic regions. Yet unclear is how handling might enhance recovery or influence other therapies such as EVs after stroke. We used the transient middle cerebral artery occlusion (MCAO) model of stroke in rats to assess how intensive handling alone, in the form of sensorimotor behavioral tests, or in combination with an intranasal treatment of EVs restored neurological function and ischemic damage. Handled rats were exposed to a battery of sensorimotor tests, including the modified Neurological Severity Score (mNSS), beam balance, corner, grid walking, forelimb placement, and cylinder tests, together with Magnetic Resonance Imaging (MRI) at 2, 7, 14, 21, and 28 days post-stroke (dps). Handled MCAO rats were also exposed to an intranasal multidose or single dose of EVs. Non-handled rats were evaluated only by mNSS and MRI at 2, 28, and 56 dps and were treated with a single intranasal dose of EVs. Our results showed that handling animals after MCAO is necessary for EVs to work at the tested dose and frequency, and that a single cumulative dose of EVs further improves the neurological function recovered during handling. These results show the importance of rehabilitation in combination with other treatments such as EVs,

**Funding:** This research was funded by the Joseph and Marie Field Laboratory for Cerebrovascular Research, and the Vickie and Jack Farber Institute for Neuroscience. The funders had no role in study design, data collection and analysis, decision to publish, or preparation of the manuscript.

**Competing interests:** The authors have declared that no competing interests exist.

and highlight how extensive behavioral testing might influence functional recovery after stroke.

## Introduction

Ischemic stroke, due to a disruption of blood supply to the brain by a clot, is a significant cause of long-term neurological disability and death in adults worldwide [1, 2]. After stroke, there is a period of spontaneous recovery where patients may see improvement in certain functions [3]. The degree and speed of spontaneous recovery after stroke is a complex process, varying between individuals depending on the severity of their deficits and areas of the brain affected [2, 4]. Rehabilitation, using multiple physical therapy interventions at the same time and considering the capabilities of each patient, is the only current therapy used to further improve both motor learning and sensory function following stroke [5]. These training interventions modulate, enhance, or accelerate spontaneous recovery after stroke by regulating plasticity mechanisms in the area surrounding the core of the injury as well as in the contralateral hemisphere and spinal cord [6–8].

Previous preclinical and clinical studies by multiple research groups, including our lab, showed that transplantation of bone marrow-mesenchymal stem cells (BM-MSCs) after stroke improves neurological recovery by modulating inflammation, promoting cell survival, and enhancing angiogenesis [9–13]. Many of these improvements are presumably due to paracrine factors packaged in extracellular vesicles (EVs) and released by BM-MSCs as a response to stroke [13–15]. BM-MSC-derived EVs contain proteins, lipids, and nucleic acids essential for cell communication that might stimulate endogenous repair processes [14]. Contrary to cell-based therapies, EVs possess a distinct advantage in their ability to readily traverse the blood-brain barrier (BBB) owing to their diminutive size [16]. Furthermore, EVs derived from allogeneic sources eliminate the necessity for immunosuppression due to their inherently low immunogenicity [17]. Most studies using EVs for the treatment of stroke are delivered through the conventional intravenous route, where one of the side effects is accumulation in the liver, lungs, kidneys, and spleen, with only moderate detection in the brain [18]. In contrast, when administrated via the non-invasive intranasal route, EVs reach the brain and target specific ischemic regions such as the striatum or the peri-infarct area [19, 20]. Previous studies have shown that intranasal administration of EVs reduces stroke volume and inflammation, prevents neuronal and oligodendrocyte cell death, and improves learning abilities after brain injury in neonatal and perinatal rats and mice [21–24]. In adult rats, a single dose of intranasal human adipose MSC-derived EVs recovered motor and cognitive symptoms, reduced the infarct volume, repaired the BBB, and stimulated angiogenesis after focal ischemia in the motor cortex [25]. A similar effect was found in a mouse model of middle cerebral artery occlusion (MCAO) exposed to intranasal BDNF-loaded EVs [20]. In most of these intranasal studies, however, sensorimotor behavioral testing was not performed [21–24] or EVs were not tested without the possible additive or synergistic effect of repetitive behavioral testing [20, 25]. Additionally, Nalamolu *et al.* [26] showed no differences between the intravenous administration of human umbilical cord blood-MSC-derived EVs and the vehicle control in ischemic rats exposed to multiple behavioral tests, finding that EVs even halted the sensorimotor recovery in some of these tests over time. On the other hand, stroke rats treated with a combination of brain-derived EVs and treadmill exercise improved neurological function compared to the sedentary group, and even further when compared to rats treated only with exercise [27]. Hence, the potential additive or synergistic impact of behavioral testing or handling on the effectiveness of EV treatment post-stroke remains inconclusive.

In preclinical studies, social housing and enriched rehabilitation (cages with multiple objects to promote forelimb use) enhanced functional recovery and increased plasticity in the contralateral cortex in a model of focal ischemia induced by endothelin-1 in the middle cerebral artery (MCA) [28]. Additionally, rats with stroke due to MCAO that exercised on a treadmill (30 minutes daily, 5 days/week for 4 weeks) showed improved neurological function, reduced infarct volume, improved corticospinal track integrity, and increased the number of synapses and plasticity-related proteins (synaptophysin and PSD-95) [27]. Although these studies demonstrate that specific rehab interventions enhance recovery in preclinical models of stroke, there is a surprising paucity of literature examining the effect that handling due to intensive exposure to sensorimotor behavioral tests has on functional recovery. Sensorimotor behavioral tests, such as the cylinder or grid walking tests, have been traditionally used to assess the degree of damage after stroke as well as the degree of recovery after a given treatment [29]. However, they are not considered a rehab intervention *per se*, even though animals are forced to perform skillful tasks multiple times throughout the experiment [29]. To our knowledge, there are no specific studies that address how intensive sensorimotor behavioral testing may enhance the therapeutic effects of intranasal administration of BM-MSC-derived EVs. Our study aims to understand how handling, in the form of intensive sensorimotor behavioral testing, affects functional recovery and injury volume after MCAO stroke in adult rats. Additionally, we seek to determine if the treatment with intranasal EVs furthers recovery or if a ceiling effect is reached after rats have been handled. Lastly, we want to determine whether EVs alone rescue the impaired neurological function or intensive somatosensory behavioral testing is needed for them to work.

## Materials and methods

### Animals

All animal work in this study was performed following the recommendations in the Guide for the Care and Use of Laboratory Animals of the National Institutes of Health. All procedures were approved by the Institutional Animal Care and Use Committee (IACUC) of Thomas Jefferson University (Protocol #00813–1). All efforts were made to minimize suffering. Due to the disparity between neurological outcomes after MCAO between males and females reported previously [30], only male rats were included in this study. Adult male Sprague-Dawley rats weighing 270–400 g, aged 8–12 weeks, were used (Envigo, #210M; RRID: RRRC_00239). Animals were maintained in the Thomas Jefferson University Animal Facility (2–3 rats housed per cage) in a 12/12-hour light/dark cycle and stable room temperature ($22 \pm 1°C$), with *ad libitum* access to food and water.

### Cohorts and experimental groups

Only animals with striatal and cortical ischemic damage after MCAO as seen by Magnetic Resonance Imaging (MRI), and a modified Neurological Severity Score (mNSS) equal to or above 9 at 2 days post-stroke (dps) were included in the study. Rats that met this inclusion criteria were randomly assigned to experimental groups (vehicle or EVs) in each cohort. Three different cohorts were established to study the effects of handling and human BM-MSC-derived EVs administered intranasally on rats after MCAO stroke: Non-handled + intranasal single dose (vehicle, *n = 12*; EVs *n = 13*), Handled + intranasal single dose (vehicle, *n = 10*; EVs, *n = 10*), Handled + intranasal multidose (vehicle, *n = 10*; EVs, *n = 10*) (S1 Table in S2 File). All animals were subjected to behavioral tests and MRI analyses following the experimental design and timeline for each experimental group. In this study, we considered handled animals those that went through intensive behavioral testing (as explained below) and non-handled those

animals that did not receive intensive behavioral testing. Each handled rat was subjected to 50–90 minutes per session (~250–450 minutes in total) of intensive behavioral testing at 2, 7, 14, 21, and 28 dps (S2 Table in S2 File). The behavioral test battery included the modified Neurological Severity Score (mNSS, S3 Table in S2 File), cylinder test, corner test, beam balance test (S4 Table in S2 File), grid walking test, and forelimb placing tests (vibrissae-evoked and proprioceptive). Non-handled rats were subjected to 15–20 minutes per session (~45–60 minutes in total) only including mNSS at 2, 28, and 56 dps. Experiments were finalized after 28 dps or 56 dps, as indicated. Details of the experimental design and timeline with the different cohorts and treatments are shown in S1a Fig in S1 File.

## Middle Cerebral Artery Occlusion (MCAO)

Ischemic injury was induced by the transient middle cerebral artery occlusion (MCAO) model using the Longa method [31]. Anesthesia was induced by 2% isoflurane, and then animals were anesthetized subcutaneously with a mixture of ketamine hydrochloride (100 mg/kg), xylazine (5 mg/kg), and acepromazine maleate (2 mg/kg). Body temperature was maintained with a heating pad at 37˚C. Before surgery, the rat's neck was shaved, and artificial tears were applied to the eyes. The animal was placed supine, and the surgical area was treated with betadine and 70% ethanol. A 2 cm midline incision on the right side was performed to access the right common carotid artery (CCA), external carotid artery (ECA), and internal carotid artery (ICA). The right CCA and ECA were ligated with silk suture (size 6/0; Fine Science Tools, #18020–60) at their most distal position. The pterygopalatine (PPA) and occipital (OA) arteries were ligated to facilitate filament insertion and stop bleeding. The superior thyroid artery (STA) was cauterized, and a microvascular clamp temporarily occluded the ICA. A loose knot was placed at the most proximal end of the ECA, near the bifurcation, and the ECA was opened with a tiny incision between the most distal ligature and slack knot. A silicone rubber-coated monofilament (Doccol, #403756PK10 or #403956PK10, RRID: SCR_015960) was chosen based on animal weight and Doccol recommendations. The filament was inserted into the ECA lumen and advanced toward the CCA. The slack knot was tightened to avoid bleeding, and the ECA was entirely cut to allow the movement of the filament toward the rostral part of the animal. The microvessel clamp was removed, and the filament was carefully advanced until resistance (~18 mm) into the ICA to occlude the middle cerebral artery (MCA). After two hours, a new ECA suture was added, and the old one near the bifurcation was loosened to remove the filament without bleeding. The CCA knot was removed before closing the midline incision, allowing total reperfusion. During the second part of the surgery to remove the filament, animals were under 1% isoflurane to maintain a sedative state. Animals received 20 ml of warmed saline, and buprenorphine (0.05mg/kg, Par Pharmaceutical), both subcutaneously, within four hours after surgery to avoid pain. Animals were monitored during the four days post-surgery. Animals were kept in a heating pad and received warmed saline (10 ml/day, subcutaneously) the first two days after surgery. Rats had free access to bacon softies (Bio-Serv, #F3580) for aid in recovery. Weights were monitored and recorded during the entire length of the experiment (S1b–S1e Fig in S1 File).

## Behavior tests

S2 Table in S2 File summarizes all behavior tests and their purpose. Rats were brought 20 min in advance to the behavioral room to allow acclimation. Each apparatus was thoroughly cleaned with 70% ethanol between rats. All tests were video recorded without identification of the experimental group to which each rat belonged, allowing the investigator to perform

analyses blinded to the treatments. Rats were tested before MCAO surgery to assess baseline (-3 dps) and ensure all animals performed without difficulties.

**Modified Neurological Severity Score (mNSS).**  This test evaluated sensorimotor activity and general neurological recovery after stroke. Our mNSS was a composite score made up of 8 independent tests [10], including the assessment of (1) postural signs, (2) gait dysfunction, (3) response to tail pull, (4) proprioceptive forelimb placing, (5) resistance to lateral displacement, (6) wire grasping strength, (7) grasping reflex, and (8) spontaneous activity [10]. Details of each independent test and their scores can be found in S3 Table in S2 File. A total score of 0 represented rats with no deficits, while 16 points was the maximum score for rats with severe deficits. Rats with an mNSS equal to or above 9 at 2 dps were included in the experimental cohorts.

**Beam balance test.**  This test assessed fine motor coordination and balance [32]. Rats were placed on a wooden beam (60 cm in length, 1.75 cm in width, 4.0 cm in height, set 90 cm off the floor, with a cushion under the beam to protect rats from falls), and their performance was scored (see S4 Table in S2 File for a description of each score). A total of 3 trials (1 min/trial) were video recorded, allowing a minimum of 30 sec between trials to avoid aversion to the test. The average of the three trials was calculated to assign the final performance for each rat.

**Corner test.**  This test assessed sensorimotor damage and postural asymmetry [33]. The apparatus was made of two boards forming a 30-degree angle. The animal was placed facing the corner, turning 180 degrees to the left or right once it reached the corner by placing one or both forepaws on the wallboards. A total of 10 trials were recorded to quantify turning preference, allowing a minimum of 30 sec between trials to avoid aversion to the test. The corner test score was calculated as follows: *CT score = [(I) / (I + C)] x 100*, where (I) is the total number of turns to the ipsilateral side (right) and (C) is the total number of turns to the contralateral side (left) after 10 trials. Sham rats turned left or right randomly, but stroke rats turned to the ipsilateral side (right) of the stroke (CT score = 100).

**Vibrissae-evoked & proprioceptive forelimb placing tests.**  These two tests assessed somatosensory and motor functions [34]. The body and limbs of the animal were held with two hands, leaving only the forelimb that was tested hanging free. For the vibrissae-evoked test, the whiskers were stimulated with the edge of the table, promoting the placement of the same-side paw on top of the table. For the proprioceptive test, the animal was handled, and the free paw was dragged to the edge of the table, eliciting a response where the rat brought the paw on top of the table. A total of 10 trials on each side were allowed, and the percentage (%) of successful placement for each paw was calculated as follows: *(number of placements of each paw / 10 trial attempts) x 100*.

**Cylinder test.**  In this test, spontaneous forelimb use assessed sensorimotor function and forelimb asymmetry. A plexiglass cylinder (20 cm in diameter, 30 cm in height) on a table was used to perform the test, together with multiple mirrors around the cylinder to monitor side-view movements. The rat was placed inside the cylinder and recorded for 2 min. The video was used to check the number of contacts from the ipsilateral paw (intact right side), contralateral paw (impaired left side), or both simultaneously. To count the touch, the animal had to return to the ground between touches and actively use the paw, making contact with all the digits and the palm. The total number of touches with left (contralateral), right (ipsilateral), or simultaneously with both (bilateral) paws was calculated. Additionally, an asymmetry score was calculated with the formula: *(ipsilateral + bilateral / 2) / (ipsilateral + contralateral + bilateral)*, where the higher (1) or lower (0) the number, the higher the asymmetry [35]. A score of 1 meant only contacts with ipsilateral paw, while 0 represented only contacts with the contralateral paw. A score of 0.5 meant that both paws were used similarly (together or

independently). Below 0.5 represented preference for the contralateral paw, and above 0.5 a preference for the ipsilateral paw.

**Grid walking test.** This test assessed fine motor function and limb coordination with accurate paw placement. Rats were placed on an elevated metal square grid ($41\times41$ cm$^2$, with each grid cell $3.5\times3.5$ cm$^2$; height: 41 cm), allowing them to walk freely for 2 min. Rats walked by placing their paws in the wires. Foot slips for each limb, and the total number of steps were recorded. The percentage (%) of errors in the contralateral (or ipsilateral) foot was calculated as follows: *(number of footfalls in each paw x 100) / total number of steps* [36, 37]. The total number of steps was measured to assess activity on the grid. Lastly, a foot fault index was also calculated as follows: *(contralateral paw faults—ipsilateral paw faults) / total steps*, where a score of 0 represented no asymmetry, a positive score represented a contralateral deficit, and a negative score represented an ipsilateral deficit [38].

## Magnetic Resonance Imaging (MRI) and analysis

A 1-Tesla M7 imaging 5 system (Aspect Imaging) was used to acquire axial 2D multi-slice T2-weighted images (T2WI) with the following parameters: voxel size: 0.16 x 0.16 x 1 mm, vertical and horizontal FOV: 30.48, # samples: 192, horizontal and vertical resolution: 0.16, number of slices: 22, inter-slice gap: 0, slice thickness:1, # excitations: 12, # phase encodings: 192. Rats were anesthetized with 2.5–3% isoflurane and placed in the rat L40D35 Handling System. Body temperature was maintained at 37˚C, and respiratory rate was continuously monitored (30–70 breaths/minute). MRI was used to assess *in vivo* the extent of ischemic damage at 2 dps, and evaluate which animals were valid for our study following our inclusion criteria (both striatal and cortical involvement). Additionally, MRI was performed at 28 dps and 56 dps in the non-handled cohort, and at 7, 14, 21, and 28 dps in the handled cohorts to assess the impact that handling and/or EVs had on ischemic damage. MRIs were always performed after all the behavioral tests to prevent changes in behavior due to anesthesia. MRI analyses were performed with VivoQuant 3.5 software (Invicro), using 15 coronal sections from rostro (Bregma ~6.70 mm) to caudal (Bregma ~-7 mm). The 3D ROI Tool was used to freely draw the contour of total ischemic damage (ischemic core) and the contour of the ipsilateral and contralateral hemispheres in each coronal section. The software automatically calculated the volumes (mm$^3$) of the ROI within the contour. The percentage (%) of damage in the ipsilateral hemisphere was calculated as follows: *(volume total ischemic damage x 100) / volume ipsilateral hemisphere*. Additionally, the percentage (%) of edematous expansion was calculated as $[(\Sigma(Ii - Ci)) / (\Sigma Ci)] \times 100$, where $Ii$ is the area of the ipsilateral hemisphere of slice i, including core of the injury, and $Ci$ is the area of the contralateral hemisphere of slice *i* [37].

## Culture of human bone marrow mesenchymal stem cells (hBM-MSCs)

Human bone marrow mesenchymal stem cells (hBM-MSCs) were purchased from Celprogren (frozen vial with ~1.2 x 10$^6$ cells; #36094–22) and plated on human mesenchymal bone marrow stem cell culture extracellular expansion matrix pre-coated T75 Flasks (Celprogen, #E36094-21-T75) following Celprogren recommendations. Cells were maintained and expanded with human (bone marrow-derived) mesenchymal stem cell complete media with serum (Celprogen, #M36094-21S) at 37˚C in a 5% CO$_2$ atmosphere. Cells were passaged before becoming confluent. Serum-free media containing the EVs was obtained from the third, fourth, and fifth passages. To that end, media with serum was removed, and cells were rinsed three times with pre-warmed Dulbecco's phosphate-buffered saline (DPBS) (Gibco, #14190136). Then, pre-warmed human (bone marrow-derived) mesenchymal stem cell

serum-free media (Celprogen, #M36094-21) was added and collected 48 hours later for EV isolation.

## Extracellular vesicles (EVs) isolation

Serum-free media containing the EVs was cleared by multistep centrifugations (500xg 10 min, 2,000xg 15 min, 10,000xg 30 min) and filtered with a 0.45 μm$^2$ filter. Then, samples were concentrated with Vivaspin® Turbo 15 centrifugal concentrator regenerated cellulose (Sartorius, #VS15TR32), following manufacturer instructions. EVs were isolated by exoEasy Maxi Kit (QIAGEN, #76064) with final elution in 1 mL of phosphate-buffered saline (PBS), following manufacturer protocol. EV isolated samples were used for EV quantification and characterization or frozen at −80˚C for further treatments.

## Particle size distribution and quantification of EVs

To analyze the particle size distribution and quantify the number of EVs, nanoparticle tracking analysis (NTA; RRID:SCR_014239) was performed using NanoSight equipment as previously described [39]. The NTA system directed a laser beam through a suspension containing the particles of interest. These particles were observed through light scattering using a traditional optical microscope aligned perpendicular to the beam axis, capturing light scattered from each particle within the field of view. The nanoparticle tracking analysis software recorded three separate 30-second video sequences of all events for subsequent examination. Using the Stokes-Einstein equation, the software tracked the Brownian motion of individual particles between frames to determine their size.

## EVs characterization by western blotting and Transmission Electron Microscopy (TEM)

As previously described, western blotting was used for protein characterization from EVs and hBM-MSCs [40]. EVs and hBM-MSCs were homogenized in ice-cold lysis buffer (NaCl 150 mM, Tris-HCl 50 mM pH 7.4, EDTA 2 mM) containing 1% Triton X-100, 0.1% SDS, 1X protease inhibitor cocktail (Sigma-Aldrich, #11836170001), 1 mM sodium orthovanadate (Sigma-Aldrich, #S6508) and 1 mM sodium fluoride (Sigma-Aldrich, #201154). Both homogenizes were sonicated three times (10 s "on" and 20 s "off") using a Diagenode Bioruptor Standard Waterbath Sonicator. Samples were spun down at 22,000×g at 4˚C. Supernatant was quantified for protein content using DC Protein Assay (Bio-Rad, #5000112). Equal amounts of protein were diluted in Laemmli buffer, boiled, and resolved using SDS-PAGE as in [41]. Proteins were then transferred onto nitrocellulose membranes (37 V overnight at 4˚C in transfer buffer containing 25 mM Tris, 192 mM glycine, 0.1% SDS, and 20% methanol) and incubated with blocking buffer (5% skim milk in Tris-buffered saline containing 0.5% Tween 20, TBST) at room temperature for 1 hour. Membranes were incubated with primary antibodies against GM130 (1:1,000, rabbit, BD Biosciences, #610822), ALIX (1:1,000, mouse, Cell Signaling, #2171), CD9 (1:1,000, Abcam, #ab92726), and CD81 (1:200, rabbit, Abcam, #ab109201) overnight at 4˚C. Membranes were rinsed three times for 10 minutes and then incubated with horseradish peroxidase (HRP)-conjugated anti-rabbit or anti-mouse antibodies (1:2,500, Cell Signaling, 7074, 7076) at room temperature for 1 hour. Development was done after 5 minutes of incubation with ELISTAR chemiluminescence substrates (Cyanagen, #XLSE077). Band density was documented using UVItec Cambridge Alliance. Morphological analysis of EVs was obtained by negative stained transmission electron microscopy (TEM). Carbon film coated 200 mesh copper grids (Electron Microscopy Sciences, #CF200-CU-50) were glow-discharged (15 mA, 0.39 mBar, 60 sec) with PELCO easiGlow Glow Discharge Cleaning System

(RRID:SCR_020396). Five μl of isolated EV sample was pipetted onto the grid and incubated for ~5min. Excess liquid was gently removed with filter paper and 2% uranyl acetate solution was added. Samples were rinsed with distilled water and allowed to dry at room temperature. Sample images were observed and taken with a FEI Tecnai T12 120kV electron microscope (RRID:SCR_022981; HV = 80kV) and a AMT XR111 CCD camera.

## Intranasal administration

All intranasal treatments were initiated at 2 dps after performing behavioral and MRI analyses. Anesthesia was induced with 2% isoflurane, rats were placed supine, and the intranasal administrations were performed using a P20 pipette. Rats were kept anesthetized with 0.5–1% isoflurane via nose cone to prevent sneezing between nostril administrations. Rats were monitored at all times to avoid respiratory depression. Animals treated with an intranasal single dose received $2.4 \times 10^9$ EVs in 200 μl (5 x 20 μl/nostril) at 2 dps. Animals treated with an intranasal multidose received 8 doses in total (twice a week for 4 weeks). Each dose contained $0.8 \times 10^9$ EVs in 120 μl (3 x 20 μl/nostril), administrating a total concentration of $2.4 \times 10^9$ EVs across the 4 weeks. Control groups received the respective total volume of vehicle (PBS) following the same procedure in the single or multidose groups.

## EVs labeling and *in vivo* tracking

We performed *in vivo* tracking of EVs by delivering fluorescence-tagged EVs (ExoGlow™ Membrane EV Labeling Kit (System Biosciences, #EXOGM600A-1)) to rats at 2 dps (*n = 4*) [40]. EVs were incubated with a mixture of reaction buffer and labeling dye for 30 min at room temperature, following the manufacturer's instructions. To remove the excess of unbound fluorescent dye, samples containing labeled EVs were passed through a PD-Spintrap G-25 microspin column (Cytiva, #GE28-9180-04) according to the manufacturer's instructions. Labeled EVs were immediately used for *in vivo* tracking in MCAO rats, receiving a single intranasal administration.

## Tissue collection, microscopy and *in vivo* quantification of EVs

Six hours after the single intranasal EV administration [40], rats were deeply anesthetized subcutaneously with a mixture of ketamine hydrochloride (100 mg/kg), xylazine (5 mg/kg), and acepromazine maleate (2 mg/kg). Euthanasia was achieved with exsanguination after transcardial perfusion with PBS (0.1 M, pH 7.4) followed by 4% paraformaldehyde (PFA). Brains were collected and kept in 4% PFA at 4°C overnight. The following day, brains were transferred to a 30% sucrose solution in 1X PBS until sank. Fixed brains were snap freeze with isopentane and stored with foil at −80°C until sectioning. A cryostat (Microm, HM505E) was used to obtain 16 μm coronal sections from the entire brain. Sections were stored in 24-well plates at −20°C in cryoprotectant until use. Five fixed coronal sections (16 μm) from the lateral ventricles to the beginning of the hippocampus (Bregma 2.20 mm, 0.70 mm, -0.4 mm, -1.30 mm, -2.56 mm), including the core of the ischemic injury, were selected. Coronal sections were rinsed three times with 1X PBS (5 minutes each). Then, sections were incubated with Hoechst 33342 (1:1,000; R&D Systems, #5117/50) in 1X PBS for 2 hours at room temperature. Sections were rinsed three times with 1X PBS (5 minutes each) and mounted on SuperFrost Plus micro slides (VWR 48311–703). Slices were covered with a 22 mm x 50 mm 1.5 thickness coverslip using ProLong Glass Antifade Mountant media (Invitrogen, #P36984) and sealed with transparent nail polish. Observation of ExoGlow-labeled EVs within the brain and Imaging was performed with a Leica Thunder Imager 3D Tissue (Leica). ExoGlow-labeled EVs were imaged in channel 594, and Hoechst in channel 405. Tile images to create a merged mosaic of the entire brain

were acquired using a 10x/0.32 NA objective (tile size 1,330.55 μm x 1,330.55 μm). Five coronal sections were analyzed per rat, and from each coronal section, one region of interest (ROI) was selected in the peri-infarct region (200 μm from the border of the injury) and the corresponding for imaging and analysis. Images were taken using a 63x/1.40–0.60 oil objective (size 211.20 μm x 211.20 μm) as z-stacks (5 steps, 6.78 μm z-size). Background was subtracted using Thunder Large Volume Computation cleaning method (adaptive strategy, refractive index 1.45700, Vectashield mounting media). ImageJ/Fiji v1.53t (RRID:SCR_002285) was used to analyze the images. A consistent threshold was applied and the average size (μm$^2$) of ExoGlow + particles was used for statistical analyses.

## Statistical analysis

Statistical analyses were performed using GraphPad Prism version 9 (RRID:SCR_002798). Results from quantification of the average size of ExoGlow+ particles were analyzed using paired t-test. MRI and behavioral results were analyzed using two-way repeated measures ANOVA for time and treatment followed by the post hoc Holm-Sidak's multiple comparisons test. Significant differences were established when the p-value was less than 0.05 ($p<0.05$). Bar graphs represent mean ± Standard Deviation (SD). Violin plot graphs represent quartiles and median. The exact number of rats used in each cohort and their treatment is specified in S1 Table in S2 File. For clarity, non-statistical results are not represented in the behavior and MRI graphs. All results (mean ± SD), including time-dependent changes, and complete statistical analysis can be found in S5 Table in S2 File.

## Results

### EVs show the expected size, markers, and morphology, and accumulate in the peri-infarct area after intranasal administration

Nanoparticle tracking analysis (NTA) revealed a peak of ~100 nm when the particle size of hBM-MSC-derived EVs was measured, which is the typical size for small vesicles such as exosomes (Fig 1a). Western blots of EVs showed the expression of characteristic EV protein markers, including ALIX and the membrane proteins CD9 and CD81 (Fig 1b). EV sample was absent of GM130, a cytoplasmic protein attached to the Golgi membrane and highly expressed in the hBM-MSC sample (Fig 1b). Transmission electron microscopy (TEM), to assess the quality and purity of the EV sample, confirmed the typical cup-shaped morphology of EVs (Fig 1c).

*In vivo* tracking of ExoGlow-labeled EVs demonstrated that EVs were present in coronal brain sections from MCAO rats. Quantification of the average size of ExoGlow+ particles showed a significant increase in the peri-infarct area (ipsilateral hemisphere) compared to the contralateral side, suggesting an accumulation of EVs in the peri-infarct area 6 hours after a single intranasal dose of EVs (2.4 x 10$^9$ EVs in 200 μl) (Fig 1d–1f).

### Handling animals after MCAO improves neurological function

To understand whether handling improves functional recovery in MCAO rats, animals were non-handled or handled (as detailed in Methods). Both non-handled and handled rats showed improved scores in mNSS at 28 dps (Fig 2a and 2b). However, there was significantly greater improvement in neurological function in the handled rat group compared to non-handled animals (Fig 2a and 2b). We used MRI imaging to assess whether handling had an impact on the volume of the injury or edema. No differences were found between handled and non-handled rats in the percentage of ipsilateral hemisphere damage (Fig 2c and 2d), total ischemic volume,

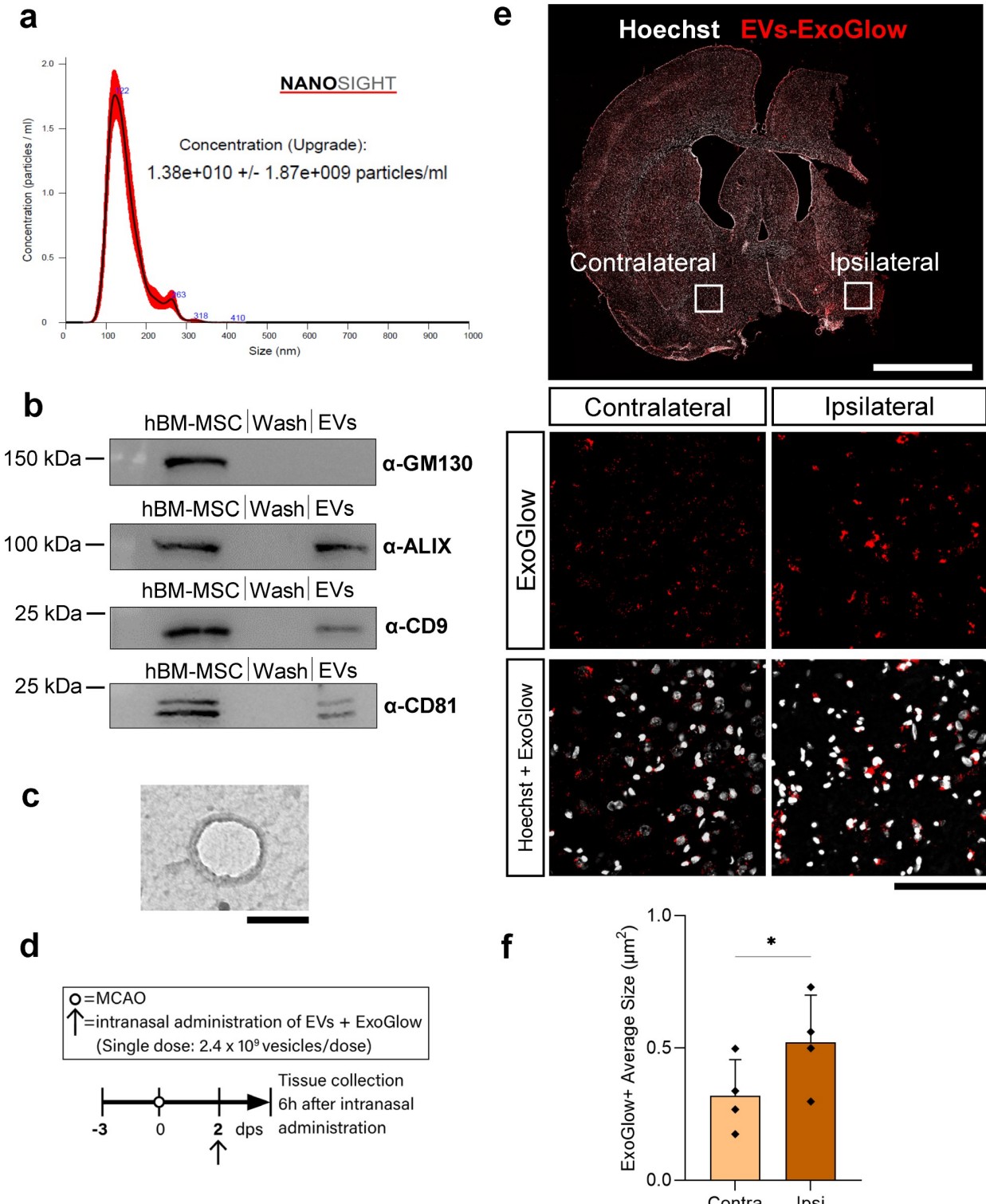

**Fig 1. EVs show the expected size, markers, and morphology, and accumulate in the peri-infarct area after intranasal administration. (a–c)** Characterization and quantification of human-bone marrow mesenchymal stem cell (hBM-MSC)-derived extracellular vesicles (EVs). **(a)** Nanosight Tracking analyses (NTA) showing the size distribution pattern of EVs. **(b)** Western blots of hBM-MSCs marker (GM130) and EVs markers (ALIX, CD9, CD81). "Wash" lane represents the negative control. Each image comes from different blots. **(c)** Representative image of EVs using transmission electron microscopy (TEM). Scale bar 100 nm. **(d–f)** *In vivo* tracking of ExoGlow-labeled EVs demonstrated that EVs reach the brain and accumulate near the stroke injury 6 hours after administration at 2 dps. **(d)** Timeline of intranasal single dose treatment of ExoGlow-

labeled EVs (~2.4 x 10$^9$ EVs in 200 µl), starting 2 dps and collecting the tissue 6 hours after. **(e)** Representative coronal section after a single intranasal administration of ExoGlow-labeled EVs (~2.4 x 10$^9$ EVs in 200 µl) at 2 dps. Hoechst indicates cell nuclei. Scale bars 4 mm and 100 µm. **(f)** Quantification of the average size of ExoGlow+ particles in the ipsilateral and contralateral hemispheres. Data was analyzed using a paired t-test (*p<0.05). Bars show mean +/− SD. Each symbol represents the average size of ExoGlow+ particles from all ROIs in each hemisphere (5 ROIs/hemisphere in each rat) of a single rat (*n* = 4).

or edema (S2a, S2b Fig in S1 File), although all measures decreased in both groups over time (Fig 2c and 2d, S2a, S2b Fig in S1 File). Taken together, these findings indicate that handled rats significantly improve neurological function compared to non-handled rats with no changes in injury volume.

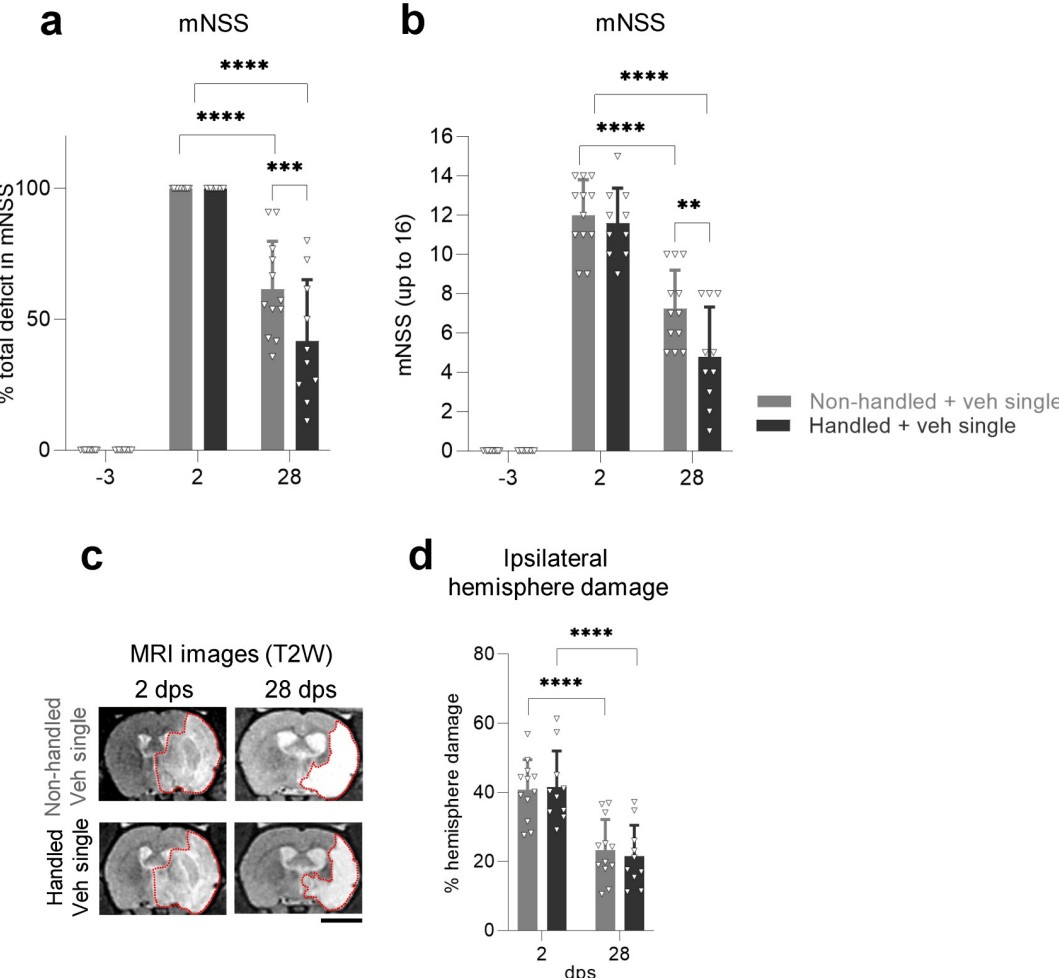

**Fig 2. Handling animals after MCAO improves neurological function. (a)** Functional recovery in non-handled and handled rats receiving a single intranasal dose of vehicle after MCAO stroke, shown as a percentage of total deficit in the modified Neurological Severity Score (mNSS) at -3 and 28 dps after normalizing the score of each animal to its score at 2 dps. **(b)** Same data as (a), showing raw mNSS results before normalization at -3, 2, or 28 dps. **(c)** Representative coronal T2-weighted MR images of non-handled and handled rats receiving a single intranasal dose of vehicle after MCAO, showing the same animal at 2 and 28 dps. Red dotted lines depict the ROI used for quantifications. Scale bar 5 mm. **(d)** Quantification of the percentage of ipsilateral hemisphere damage using MR images of non-handled or handled MCAO rats receiving a single intranasal dose of vehicle after MCAO, showing the same animal at 2 and 28 dps. All data was analyzed by two-way repeated measures ANOVA followed by the post hoc Holm-Sidak's multiple comparisons test (**p<0.005, ***p<0.0005, ****p<0.0001). Bars show mean +/− SD. Each symbol represents a single rat: non-handled + veh single (*n* = 12), handled + veh single (*n* = 10).

## Intranasal multidose of EVs does not further improve neurological function in handled rats

We next asked whether a multidose of EVs ($0.8 \times 10^9$ EVs in 120 μl, twice a week for 4 weeks) could enhance the recovery of handled rats. We found no difference in mNSS at 28 dps in handled rats treated with multidose EVs versus handled rats that received a multidose of vehicle (Fig 3a and 3b). Both groups showed improvement in mNSS over time, starting their recovery at 7 dps and plateauing at 14 dps (Fig 3a and 3b). When other behavioral tests were analyzed, no significant differences were found with multidose EVs compared to vehicle (Fig 3c–3h, S3a–S3h Fig in S1 File). We found a decrease over time in the percentage of ischemic volume, total ischemic volume, and edema in the handled group, with no additional reduction with the multidose EV treatment (Fig 3i and 3j, S2c, S2d Fig in S1 File). In sum, these results show that multidose EV administration given at the tested dose does not improve anatomical or functional recovery of handled rats after MCAO.

## A cumulative single intranasal dose of EVs improves neurological function compared to vehicle controls

Next, we tested if a single intranasal dose of $2.4 \times 10^9$ EVs in 200 μl (equivalent to the total EVs that rats received as a multidose) at 2 dps further improved recovery in handled rats. As we observed before, handled rats improved their mNSS over time, starting their recovery at 7 dps (Fig 4a and 4b). Rats that received the single intranasal dose of EVs showed a modest but significant improvement in mNSS at 28 dps (Fig 4a). Likewise, handled animals treated with a single dose of EVs performed significantly better than controls in the beam balance test at 28 dps, showing a mild improvement (Fig 4c). No differences were found between treatments (single dose of EV versus vehicle) in the corner test (Fig 4d), vibrissae-evoked placement test (Fig 4e), cylinder test (Fig 4f, S4a–S4c Fig in S1 File), or grid walking test (Fig 4g, S4d–S4f Fig in S1 File). To note, handled animals treated with a single dose of EVs performed significantly better than controls in the proprioceptive placement test at 14 dps (Fig 4h). As in other conditions tested, we did not observe reductions in the percentage of ipsilateral hemisphere damage (Fig 4i and 4j), total ischemic volume, and edema (S2e, S2f Fig in S1 File) at 28 dps. Additionally, when handled rats treated with EVs single dose were compared to handled rats treated with EVs multidose, no differences were found between these two groups in the mNSS (S5a, S5b Fig in S1 File), as well as the percentage of ipsilateral hemisphere damage (S5c Fig in S1 File), total ischemic volume (S5d Fig in S1 File), and edema (S5e Fig in S1 File) at any time point. Taken together, these results demonstrate that an intranasal administration of a single dose of EVs at 2 dps at a concentration of $2.4 \times 10^9$ EVs in 200 μl modestly enhances behavioral recovery in handled rats after MCAO without changes in stroke volume compared to vehicle controls.

## Single dose of EVs alone does not aid neurological recovery

Last, we asked if a single dose of EVs ($2.4 \times 10^9$ EVs in 200 μl) without handling was sufficient to improve functional recovery after MCAO. We demonstrated that a single intranasal dose of EVs alone at the tested dose does not improve the mNSS in non-handled animals at 28 or 56 dps (Fig 5a and 5b). There was a time-dependent improvement of the mNSS (Fig 5a and 5b) and reduction in the percentage of ischemic volume (Fig 5c and 5d), total ischemic volume, and edema (S2g, S2h Fig in S1 File), without differences between treatment groups. Altogether, this shows that a single cumulative dose of EVs at the given dose is more effective at improving recovery when used in combination with handling.

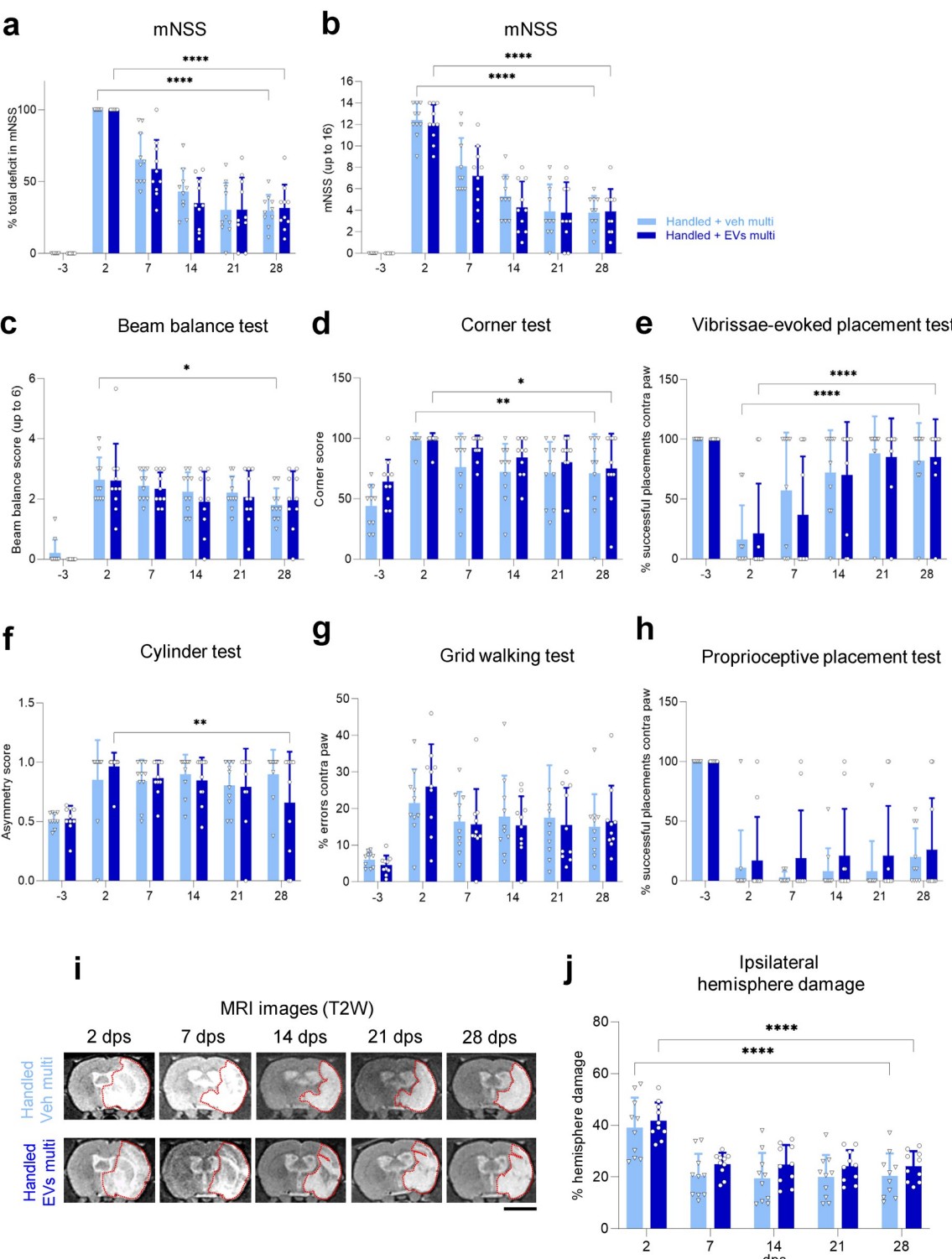

**Fig 3. Intranasal multidose of EVs does not further improve neurological function in handled rats. (a)** Functional recovery in handled rats receiving a multidose of vehicle or EVs after MCAO stroke, shown as a percentage in total deficit in the modified Neurological Severity Score (mNSS) at -3, 7, 14, 21, and 28 dps after normalizing the score of each animal to its score at 2 dps. **(b)** Same data as (a), showing raw mNSS results before normalization at -3, 2, 7, 14, 21, and 28 dps. **(c–h)** Handled rats treated with a multidose of vehicle or EVs were tested at -3, 2, 7, 14, 21, and 28 dps on **(c)** beam balance test, **(d)** corner test, **(e)** vibrissae-evoked forelimb placing test, **(f)** cylinder test, **(g)** walking grid test, and **(h)** proprioceptive forelimb placing test. **(i)** Representative coronal T2-weighted

MR images of handled rats treated with a multidose of vehicle or EVs after MCAO stroke, showing the same animal at 2, 7, 14, 21, and 28 dps. Red dotted lines depict the ROI used for quantifications. Scale bar 5 mm. **(j)** Quantification of the percentage of ipsilateral hemisphere damage using MR images at 2, 7, 14, 21, and 28 dps of handled rats treated with a multidose of vehicle or EVs after MCAO stroke. All data was analyzed by two-way repeated measures ANOVA followed by the post hoc Holm-Sidak's multiple comparisons test (*p<0.05, **p<0.005, ****p<0.0001). Bars show mean +/− SD. Each symbol represents a single rat: handled + veh multi (*n = 10*), handled + EVs multi (*n = 10*).

## Discussion

In this study, we tested whether intranasal EVs and handling (i.e., intensive behavioral testing), either alone or in combination, promoted recovery after MCAO in rats. Our study evaluated neurological recovery using the mNSS and other behavioral tests, and ischemic injury progression using MRI after intensive handling and/or EVs treatment. We found that an intranasal single dose of $2.4 \times 10^9$ EVs in 200 µl modestly, but significantly, enhanced the functional recovery observed when rats were handled during intensive behavioral testing. Importantly, we identified that EVs alone, at the tested dose and frequency, did not improve behavioral recovery after stroke, suggesting a synergistic effect between handling and EV administration, in which handling was needed for EVs to show a therapeutic effect.

Previous studies have established the relevance of intensive exercise rehabilitation and repetitive task-specific training (e.g., treadmill, pellet reaching) in recovery after stroke [27, 42–46]. However, the contribution of sensorimotor behavioral testing to recovery has been overlooked, even though these tests are typically used by stroke researchers to assess improvement and can include skilled-related tasks [29]. Our results indicate that weekly exposure to seven different behavioral tests improves neurological recovery in the mNSS at 28 dps compared to non-handled rats. Handled rats started their recovery in the mNSS at 7 dps, plateauing around 14 or 21 dps depending on the group. Although we did not study the lack of handling on the temporal recovery of mNSS, previous data suggests that MCAO rats exposed only to mNSS progressively improve spontaneously over time, obtaining a recovery at 28 dps similar to our results in non-handled rats [47, 48]. Functional recovery depends on the ability of the brain to undertake the lost functions in certain areas after ischemic damage. Although size and location of stroke play an essential role in this process, changes in the environment with behavioral interventions can modulate this sensorimotor recovery [49, 50]. For instance, an enriched environment with social interactions, voluntary physical activity, and exposure to novel objects positively impact functional recovery after stroke in rodents [51–55]. These results highlight the importance of considering and not underestimating the role of behavioral testing as an unintended intervention with the potential to influence functional results in control and treatment groups after stroke.

Also of great interest in our studies was the fact that EVs given as a single dose ($2.4 \times 10^9$ EVs in 200 µl), but not when subdivided into multiple doses over time, moderately enhanced the neurological recovery seen with handling. Long periods of repeated exposure to isoflurane may be an influencing factor in recovery [56–58]. In our study, to bypass this potential problem, we used appropriate controls and shorter periods of isoflurane exposure compared to previous research (e.g., 1 hour of isoflurane just after removing the filament in the MCAO model and starting reperfusion). Moderate recovery with a single dose was specifically seen in the mNSS and in the beam balance test, which suggests improved motor coordination and balance. Although this might suggest that a ceiling effect is not reached after physical handling and highlights the promise of physical rehabilitation used in combination with EV treatment (or other pharmacological or cell-derived treatments), the lack of sensorimotor improvement when directly comparing the single and multidose of EVs at 28 dps indicates that the main

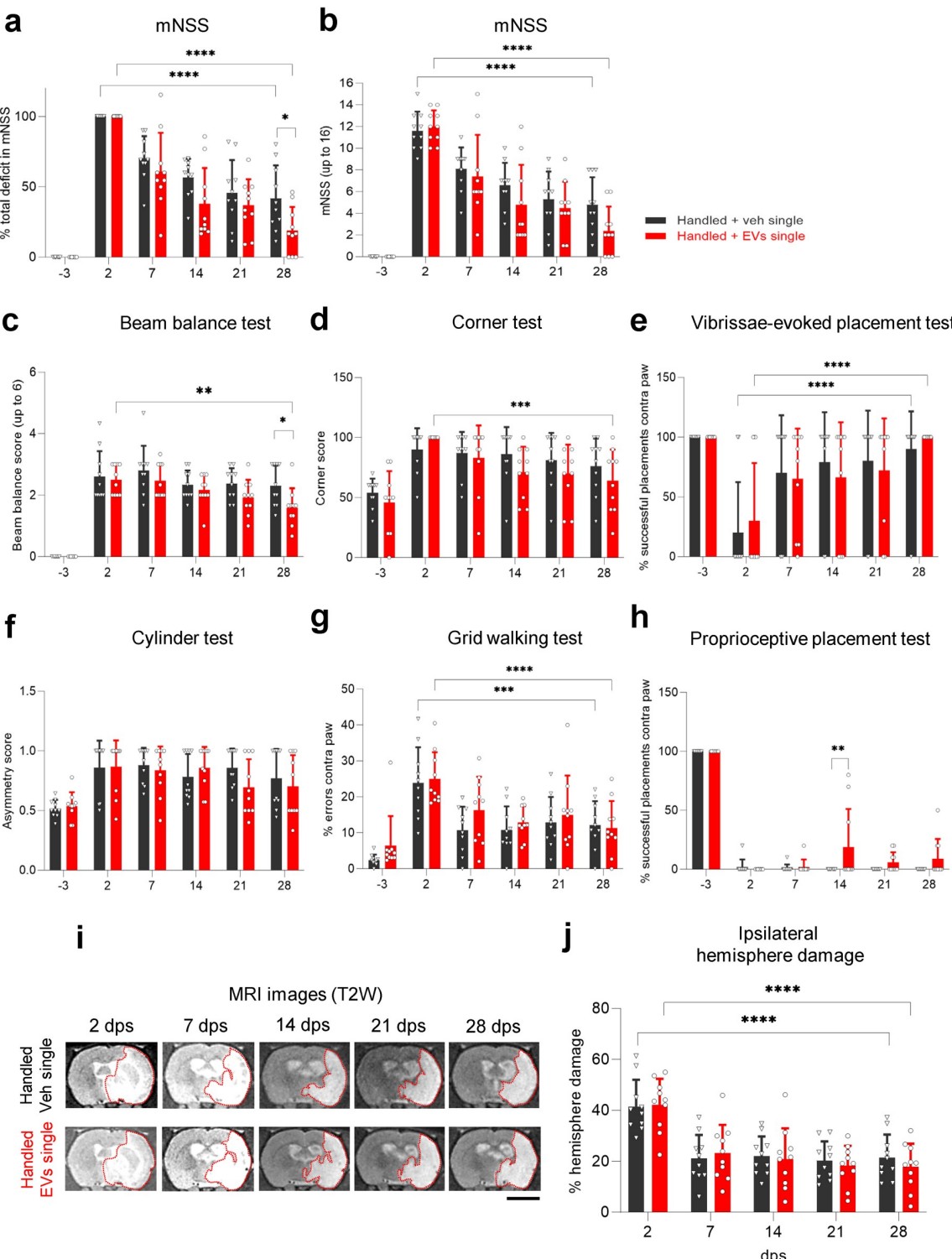

**Fig 4. A cumulative single intranasal dose of EVs improves neurological function compared to vehicle controls. (a)** Functional recovery in handled rats receiving a single dose of vehicle or EVs after MCAO stroke, shown as a percentage in total deficit in the modified Neurological Severity Score (mNSS) at -3, 7, 14, 21, and 28 dps after normalizing the score of each animal to its score at 2 dps. **(b)** Same data as (a), showing raw mNSS results before normalization at -3, 2, 7, 14, 21, and 28 dps. **(c–h)** Handled rats treated with a single dose of vehicle or EVs were tested at -3, 2, 7, 14, 21, and 28 dps on **(c)** beam balance test, **(d)** corner test, **(e)** vibrissae-evoked forelimb placing test, **(f)** cylinder test, **(g)** walking grid test, and **(h)** proprioceptive forelimb placing test. **(i)** Representative coronal

T2-weighted MR images of handled rats treated with a single dose of vehicle or EVs after MCAO stroke, showing the same animal at 2, 7, 14, 21, and 28 dps. Red dotted lines depict the ROI used for quantifications. Scale bar 5 mm. **(j)** Quantification of the percentage of ipsilateral hemisphere damage using MR images at 2, 7, 14, 21, and 28 dps of handled rats treated with a single dose of vehicle or EVs after MCAO stroke. All data was analyzed by two-way repeated measures ANOVA followed by the post hoc Holm-Sidak's multiple comparisons test (*p<0.05, **p<0.005, ***p<0.0005, ****p<0.0001). Bars show mean +/− SD. Each symbol represents a single rat: handled + veh single (*n = 10*), handled + EVs single (*n = 10*).

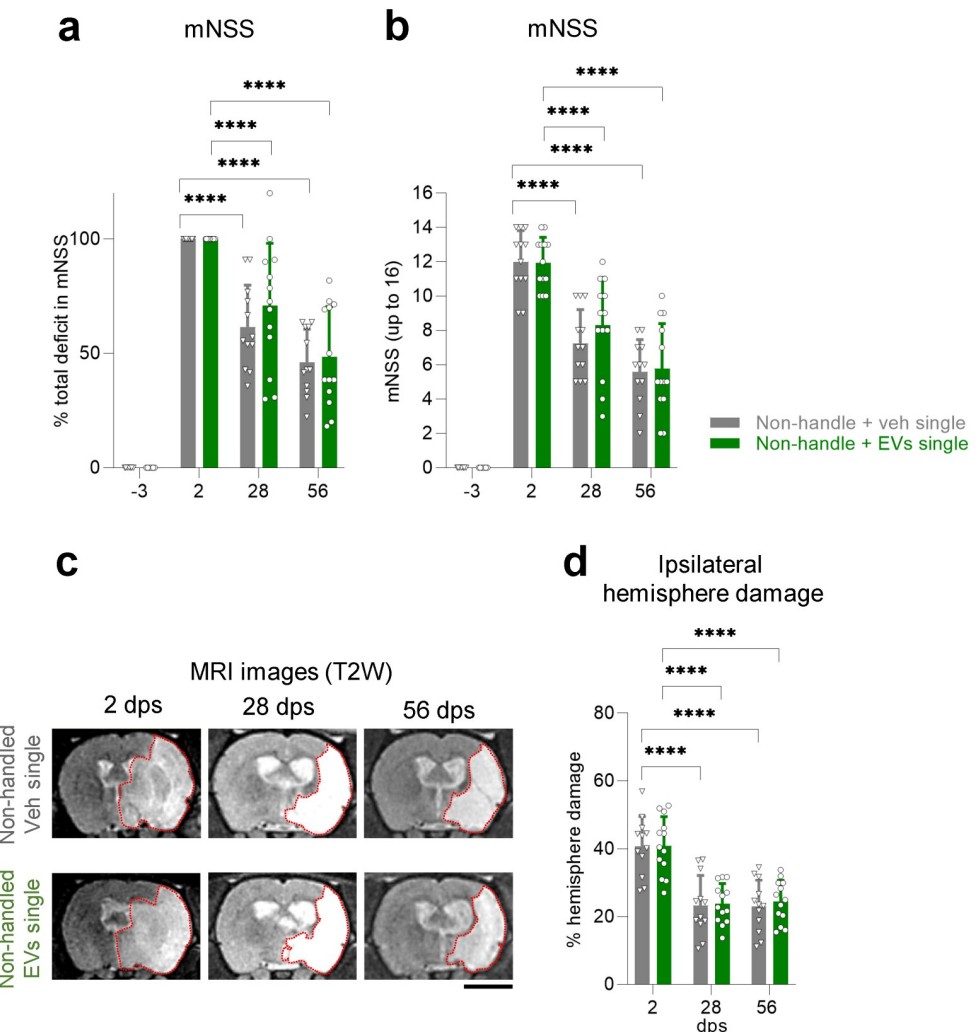

**Fig 5. Single dose of EVs alone does not aid neurological recovery. (a)** Functional recovery in non-handled rats treated with a single intranasal dose of vehicle or EVs after MCAO stroke, shown as a percentage in total deficit in the modified Neurological Severity Score (mNSS) at -3, 28 and 56 dps after normalizing the score of each animal to its score at 2 dps. **(b)** Same data as (a), showing raw mNSS results before normalization at -3, 2, 28, or 56 dps. **(c)** Representative coronal T2-weighted MR images of non-handled rats treated with a single intranasal dose of vehicle or EVs after MCAO, showing the same animal at 2, 28, and 56 dps. Red dotted lines depict the ROI used for quantifications. Scale bar 5 mm. **(d)** Quantification of the percentage of ipsilateral hemisphere damage using MR images of non-handled rats treated with a single intranasal dose of vehicle or EVs after MCAO at 2, 28, or 56 dps. All data was analyzed by two-way repeated measures ANOVA followed by the post hoc Holm-Sidak's multiple comparisons test (****p<0.0001). Bars show mean +/− SD. Each symbol represents a single rat: non-handled + veh single (*n = 12*), non-handled + EVs single (*n = 13*).

efficacy is derived from handling. This is especially relevant when considering clinical studies, as most patients undergo physical rehabilitation [4]. We initially decided on a multidose regimen to maintain a more continual source of EVs in an attempt to mimic the effects of BM-MSC grafts in the brain after stroke [10]. The most plausible explanation for the lack of efficacy in MCAO rats treated with multidose EVs is that the concentration of EVs at any given time was insufficient even though the dose we used was based on a previously published study [40]. This could be critically important at 2 dps, considering that a minimum dose might be essential to trigger recovery mechanisms. It is possible that functional and neuroanatomical recovery will improve when a higher intranasal dose is given, as a single or multidose. In fact, a recent article has shown an improvement in behavioral function and infarct volume after MCAO in mice using an intranasal multidose of EVs that was approximately ten times higher in each dose than our treatment [20]. Numerous publications have utilized EVs in a variety of models of ischemia [20, 59–69], but surprisingly, an ideal dose of EVs has yet to be determined. This is probably due to the many factors that come into play when assessing and reporting the efficacy of EVs in stroke [70]. For example, there is no well-established common method to report the quantity of EVs used in each study [71], making it difficult to compare between doses.

Likewise, the source of EVs is critically important to efficacy [70]. In our work, we used BM-MSCs-derived EVs, which have been shown to have a high regeneration ability compared to EVs derived from adipose tissue- or umbilical cord-MSCs [72]. Specifically, MCAO functional recovery after BM-MSCs-derived EVs treatment (using different routes of administration) has been linked to complex and multifactorial repair mechanisms, such as a reduction of microglia and macrophage infiltration, increase of angiogenesis and neurogenesis, decrease of neuronal apoptosis, or enhance of synaptic plasticity [63, 65, 69]. Additionally, MSCs, which interact continuously with ischemic damaged tissue when placed directly in the brain or body after stroke, may release EVs containing more relevant molecules to the treatment of stroke [59]. In our study, we used EVs obtained from healthy human BM-MSCs and cultured in non-ischemic conditions which may be different in their molecular content compared to EVs released from BM-MSCs exposed to ischemic signals. Thus, thorough characterization of EVs derived from cellular sources exposed to different conditions will be an important next step in advancing the field and understanding potential treatments.

Additionally, the route of administration of EVs is of paramount importance as it determines the degree to which they cross the BBB and reach the brain [70]. Other authors have shown functional improvement after BM-MSCs-derived EV injection using multiple sensorimotor tests via intravenous injection [60, 63–65, 68, 69], one of the most common routes of EV administration, together with intra-cerebral and intra-arterial routes [59, 61, 62, 66, 67]. These routes are more invasive and, therefore, less attractive for the clinic than the non-invasive intranasal route [73], which could be easily applied to human patients after stroke. In our study, we confirmed that intranasal EVs, given as a cumulative single dose, reach the brain and accumulate in the peri-infarct area six hours after their administration at 2 dps, as reported in previous publications [19–21, 25, 74, 75]. We did not confirm the long-term presence of EVs in the brain after 28 dps due to characteristics of the fluorescent dye that can lead to misleading results (e.g., aggregation, dye half-life impacting detection overtime, phagocytosis of labeled EVs, dye uptake after EV release) [76, 77]. Nevertheless, other authors have shown, using gold nanoparticles to label EVs and computed tomography (CT) [19, 74], that intranasally administrated EVs are retained for at least 96 hours in the brain of a mouse model of focal stroke (ETH1-1 injection) around the ischemic region.

To our knowledge, our study is the first one to show a modest but significant functional recovery in the mNSS and beam balance test in adult rats with MCAO strokes, where the striatum and cortex are largely affected, when intranasal EV treatment is combined with handling. The effect of intranasal EV treatment was previously shown in neonatal and perinatal rodents [21, 23, 24], in adult rodents using endothelin-1 or thermocoagulation to generate smaller focal injuries [19, 25], and MCAO in mice [20]. Depending on the size and location of the injury, affecting different brain areas that control specific skilled movements, some regions might be easier to restore than others after handling interventions and/or EV treatments [78, 79]. For instance, specific areas from the motor (M1) and somatosensory (S1) cortex form a mosaic that highly controls the forelimb, hindlimb, and trunk, regulating single movements of the shoulder, elbow, forearm, wrist, and digits, as well as complex movements of the forearm, such as grasping or reaching [80–85]. The size of the damaged brain area may determine the efficacy of handling and/or intranasal EV treatment. It will be interesting to test if the same treatments are more effective in smaller focal ischemic lesions. The recovery will also depend on the animal's ability to learn that specific task again, using alternative brain areas to develop compensatory mechanisms [46, 85, 86]. Training interventions will modulate, enhance, or accelerate this recovery by regulating time-sensitive plasticity [87]. The lack of this sensorimotor learning process may produce maladaptive plasticity, affecting neurological improvement [6, 49, 87]. Certain behaviors might never be recoverable again or need more skill-focused tasks or repetition than others. This could explain why certain behaviors (mNSS and beam balance test) and not others improved in our study.

Lastly, we consistently saw that the percentage of ischemic volume, total infarct volume, and edema improved over time but not with handling or EV treatment. In our work, we performed *in vivo* longitudinal examinations of the ischemic core using MRI, observing the size of the injury in the same rat over time without affecting the integrity of the tissue. Like Otero-Ortega *et al.* [64], we saw no changes in lesion size using MRI with any treatment, even though motor recovery existed. In their publication, they attributed the motor recovery after EVs administration to an increase of axonal connectivity and repair of the white matter, seen with axial diffusivity by Diffusion Tensor Imaging (DTI). Although traditionally there has been a general idea that a reduction of ischemic damage is needed to obtain sensorimotor recovery, it is important to note that the published literature about the relationship between motor recovery and reduction of lesion volume remains conflicted. Some studies show an improvement after behavioral interventions and/or EV administration [27, 45, 46, 59, 62, 63], while others observed no differences in ischemic lesion size [68, 69]. These conflicting results may be attributed to the use of different techniques to assess injury volume [27, 45, 46, 59–61, 64, 65]. This suggests that a better understanding of how neuroanatomical changes in the core of the injury correlate to functional recovery is needed, highlighting the role of other mechanisms such as the formation of new sprouting fibers and synapses outside the ischemic core [7, 49].

## Conclusions

Intensive sensorimotor behavioral tests, used to assess recovery in the field of stroke, underpin the success of EVs after MCAO in rats. Thus, it is important to consider this physical intervention during pharmacological or cell-based treatments, as these might not work in the absence of handling. Follow-up research should focus on the repair mechanisms by which handling and EVs exert their benefits. This, in turn, might shed some light on our understanding of injury and recovery after stroke, and aid in the selection of treatments that improve damage-specific sensorimotor behaviors.

## Supporting information

**S1 File.**
(PDF)

**S2 File.**
(ZIP)

**S1 Raw image. Original uncropped blots and TEM image from Fig 1b and 1c.**
(PDF)

## Acknowledgments

We thank the Joseph and Marie Field for the kind gift of the MRI system; Max I. Myers and Kevin J. Hines for their training to perform the MCAO model; Gabrielle Spagnuolo and Eileen Collyer for their training and advice to perform the behavioral tests; Zuzana Nichtova for her impeccable assistance with TEM; and Andrew Gray for his technical support. Diagrams in Fig 1d and S1a Fig in S1 File were created using Adobe InDesing (RRID:SCR_021799).

## Author Contributions

**Conceptualization:** Claudio Grassi, Lorraine Iacovitti, Elena Blanco-Suarez.

**Data curation:** Yolanda Gomez-Galvez.

**Formal analysis:** Yolanda Gomez-Galvez.

**Funding acquisition:** Claudio Grassi, Lorraine Iacovitti, Elena Blanco-Suarez.

**Investigation:** Yolanda Gomez-Galvez, Malvika Gupta, Mandeep Kaur, Salvatore Fusco, Maria Vittoria Podda.

**Methodology:** Yolanda Gomez-Galvez, Malvika Gupta, Mandeep Kaur, Salvatore Fusco, Maria Vittoria Podda.

**Resources:** Claudio Grassi, Amit K. Srivastava, Lorraine Iacovitti, Elena Blanco-Suarez.

**Supervision:** Claudio Grassi, Amit K. Srivastava, Lorraine Iacovitti, Elena Blanco-Suarez.

**Visualization:** Yolanda Gomez-Galvez.

**Writing – original draft:** Yolanda Gomez-Galvez, Lorraine Iacovitti, Elena Blanco-Suarez.

**Writing – review & editing:** Yolanda Gomez-Galvez, Claudio Grassi, Amit K. Srivastava, Lorraine Iacovitti, Elena Blanco-Suarez.

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
