## [Decision Letter · Decision Letter 0]

8 May 2024

PONE-D-24-14380Recovery after human bone marrow mesenchymal stem cells (hMB-MSCs)-derived extracellular vesicles (EVs) treatment in post-MCAO rats requires handling associated with repeated behavioral testing.PLOS ONE

Dear Dr. Blanco-Suarez,

Thank you for submitting your manuscript to PLOS ONE. After careful consideration, we feel that it has merit but does not fully meet PLOS ONE’s publication criteria as it currently stands. Therefore, we invite you to submit a revised version of the manuscript that addresses the points raised during the review process.

We look forward to receiving your revised manuscript.

Kind regards,

Hansen Chen

Academic Editor

PLOS ONE

Journal Requirements:

"The authors declare no competing interests."

5. We note that Figure 1 in your submission contain copyrighted image. All PLOS content is published under the Creative Commons Attribution License (CC BY 4.0), which means that the manuscript, images, and Supporting Information files will be freely available online, and any third party is permitted to access, download, copy, distribute, and use these materials in any way, even commercially, with proper attribution. For more information, see our copyright guidelines: http://journals.plos.org/plosone/s/licenses-and-copyright.

Additional Editor Comments:

Dear Authors,

Please address the reviewers’ comments. Additionally, I have included my own comments below for your consideration:

1. Please specify the number of rats housed per cage in the Methods section.

2. Clarify whether the rats were randomized when assigned to groups, and indicate whether the surgeons and investigators conducting behavioral tests were blinded to the treatments.

3. In the Methods section, it is mentioned that the rats received 20ml of warm saline. Could you specify whether this was administered via intraperitoneal injection?

4. The repeated treatment group also involves repeated exposure to isoflurane, which might affect the recovery process. Please discuss this potential impact in your manuscript.

Reviewers' comments:

Reviewer's Responses to Questions

**Comments to the Author**

1. Is the manuscript technically sound, and do the data support the conclusions?

Reviewer #1: Partly

Reviewer #2: Yes

Reviewer #3: Partly

2. Has the statistical analysis been performed appropriately and rigorously? 

Reviewer #1: No

Reviewer #2: Yes

Reviewer #3: No

3. Have the authors made all data underlying the findings in their manuscript fully available?

Reviewer #1: No

Reviewer #2: Yes

Reviewer #3: Yes

4. Is the manuscript presented in an intelligible fashion and written in standard English?

Reviewer #1: Yes

Reviewer #2: Yes

Reviewer #3: Yes

5. Review Comments to the Author

Reviewer #1: The authors administered extracellular vesicles (EVs) derived from bone marrow mesenchymal stem cells (BM-MSCs) into the ischemic brain of MCAO models to enhance functional recovery. They evaluated the efficacy of single versus multiple doses using behavioral tests and assessed injury size over time. While some data supports their conclusions, additional information is needed before acceptance of the manuscript.

Major Concerns:

The MRI images do not correlate consistently with the quantification of brain damage depicted in Figures 2, 3, and 4.

The ischemic brain typically experiences edema followed by chronic inflammation. Behavioral tests concluding on day 28 are insufficient to confirm functional recovery in the animals.

How did EVs affect the damaged brain—by reducing inflammation or providing nutritional support? What differentiates single-dose from multi-dose treatments? The authors should explain the underlying mechanisms supporting their findings.

The authors should specify the number of animals used in behavioral tests, given the substantial standard deviation observed in Figures 3 and 4.

Minor Revisions:

In Figure 1b, molecular weight ladders are not included in the Western blot (WB) images.

Reviewer #2: This manuscript carefully investigates the effects of handling and/or intranasal EVs on the recovery after MCAO in rats, providing solid evidence that intensive behavior testing alone improves recovery after stroke, which could be enhanced after intranasal single dose of EVs. The authors have done rigorous experiments addressing this, while some concerns are needed to be addressed to fully clarify the statement.

Major points:

1. Fig. 2: effect of handling on neurological function after MCAO

The authors have shown that intensive behavior tests improve scores in mNSS as 28 dps. This is quite meaningful to the field since behavior tests have been typically used to assess performance after stroke and the contribution to stroke recovery has been overlooked. To further address the temporal effect of handling, it would be interesting to conduct these experiments at different timepoints, for example D7, D14, D21 dps like Fig. 3-4.

2. Fig. 3-4: effect of EVs (single and multiple doses) on neurological function after MCAO

The authors further explored whether intranasal EVs could enhance the handling-mediated neurological improvement after stroke at different time points, and found that single dose, instead of multiple doses showed synergistic effect, which might result from the lack of efficacy treated with multidose EVs according to the authors’ discussion. It would be important to address some concerns to further support the current statement.

EVs presence in the brain? The authors have nicely observed the presence of EVs 6 hours after a single dose (2.4 x 10^9 vesicles). How about the time course? It would be interesting to see if these EVs persists in the brain since most improvement after a single dose were observed in D28 dps (Fig. 4a-c). In addition, does EVs present in brains from MCAO rats after the lower multiple doses (0.8 x 10^9 vesicles)? A lower presence of EVs may explain the undetectable effects in stroke (Fig. 3).

Comparison between single dose and multiple doses of EVs on neurological function after MCAO? The authors have done separate experiments to compare the effect of single (Fig. 4) or multiple doses (Fig. 3) of EVs to handling alone on neurological function. Considering the baseline of most behavior tests may vary, it would be interesting to see how the dose of EVs may affect the handling-mediated neurological function in one set of experiment, or at least discusse in an integrated data from Fig. 3-4.

Statistics analysis of EVs on neurological function after MCAO? The authors have conducted 2-way ANOVA to include both time and treatment (veh or EVs) to assess if there is a significant improvement, and nicely provided the raw data for the comparison in all individual days in the supplement files. However, it would be necessary to include the P value for a row and column factor (time and treatment here) in a 2-way ANOVA test. This would be more relevant to suggest if an overall improvement exist over time and/or after EVs treatment, in addition to the improvement in specific days.

3. MRI analysis on the volume of the injury or edema

The authors have shown that either handling or EVs treatment had no effect on changes in injury volume after MCAO in rats, in contrast to the neurological function (Fig. 2-5). In Fig. 2c-d, less ipsilateral hemisphere damages were observed at D28 dps, while the representative images didn’t show a less bright volume. It would be necessary to clarify how the damage area were calculated in the Methods, and make sure to include representative images. Similar in Fig. 3-5.

Minor points:

1. Cohorts and experimental groups in Methods:

The authors described that three different cohorts were established (line 133-135). In fact, the authors have done 5 groups rigorously to evaluate the effects of both handling and EVs treatment on stroke recovery (Table S1). It would be nice to also include the group information in Methods to make it easier to follow.

Reviewer #3: This is a well written manuscript, I feel comfortable reading it. However, there are some major concerns about the figures and the statistical analysis.

1. Figure 1B. The western blot should include the markers showing the MW of the band.

2. Figure 1E. The Hoechest staining seems to be out of focus in the contralateral image, the authors needs a better image and do the quantification of the percentage of ExoGlow+ cells.

3. Figure 2D. In the test, the authors wrote "No differences were found between handled and non-handled rats in the percentage of ipsilateral hemisphere damage." The point is "no difference". Therefore, the authors should compare "handled" vs "non-handled", not "2dps" vs "28dps" in quantification.

4. Figure 2D. The authors should discuss why handling improve neurological function, but not the ipsilateral hemisphere damage. What are the link between these two tests and what could be the potential explanation that two assays shows different results.

5. Figure 3. The authors are comparing handling vs handling+multi EVs in this figure, but in every panel the authors labels significant difference between "2dps and 28dps", but not "handling vs handling+multi EVs" this is quite misleading to readers.

6. PLOS authors have the option to publish the peer review history of their article (what does this mean?). If published, this will include your full peer review and any attached files.

Reviewer #1: No

Reviewer #2: No

Reviewer #3: **Yes: **Chenjie Pan

---

## [Author Response · Author response to Decision Letter 0]

12 Jul 2024

Editor Comments

1. Please specify the number of rats housed per cage in the Methods section.

The number of rats housed per cage is now specified in the Methods section (line 127).

2. Clarify whether the rats were randomized when assigned to groups, and indicate whether the surgeons and investigators conducting behavioral tests were blinded to the treatments.

As suggested, more clarification has been added to the randomization (line 134) and blinding (lines 183-185) part of this study in the Methods section. 

3. In the Methods section, it is mentioned that the rats received 20ml of warm saline. Could you specify whether this was administered via intraperitoneal injection?

Rats received warm saline subcutaneously, and this has been added to the text (lines 175 and 177). 

4. The repeated treatment group also involves repeated exposure to isoflurane, which might affect the recovery process. Please discuss this potential impact in your manuscript.

This is a very interesting and valid point previously demonstrated in multiple publications using the MCAO model in rats (Kim et al., 2015, PMID: 31024240; Wang et al., 2016, PMID: 26898453; Peng et al., 2019, PMID: 31024240). All our experimental groups include vehicle controls, except when “Non-handled + veh single” was compared to “Handled + veh single” (Fig 2). In this case, the “Handled” group received an extra ~15 min/day of exposure to isoflurane at 7, 14, and 21 dps due to MRIs. Our experimental animals received lower doses of isoflurane compared to other studies (e.g., 1h of isoflurane just after removing the filament and starting reperfusion in references) (Kim et al, 2015, PMID: 31024240; Wang et al, 2016, PMID: 26898453; Peng et al, 2019, PMID: 31024240). Thus, according to previous reports, we believe our short isoflurane exposure is insufficient to enhance recovery. As suggested by the reviewer, we acknowledge and discuss this possibility in the manuscript (lines 567-571). 

Reviewer #1 

The authors administered extracellular vesicles (EVs) derived from bone marrow mesenchymal stem cells (BM-MSCs) into the ischemic brain of MCAO models to enhance functional recovery. They evaluated the efficacy of single versus multiple doses using behavioral tests and assessed injury size over time. While some data supports their conclusions, additional information is needed before acceptance of the manuscript.

Major Concerns:

1. The MRI images do not correlate consistently with the quantification of brain damage depicted in Figs 2, 3, and 4.

We have replaced some of the MRI images in Figs 2–5. We have added a red dotted line to delineate the ischemic core. To clarify the MRI analysis, more details have been added to the methods section (lines 255-264) and Fig legends. 

2. The ischemic brain typically experiences edema followed by chronic inflammation. Behavioral tests concluding on day 28 are insufficient to confirm functional recovery in the animals.

This is a valid concern especially because our treatments could have affected edema. There is a significant reduction in the % of the edematous expansion at 28 dps compared to 2 dps in all four groups analyzed, starting at 7 dps and stabilizing between 14 and 28 dps (Figs S2b, S2d, S2f, and S2h; Table S5). This suggests that all groups resolved the edema by 28 dps, independently of their treatment (handling or EVs). This, of course, affects functional recovery in the animals, but it does not explain the differences found between treatment groups. Additionally, the time point of 28 dps has been previously used in many other publications to assess functional recovery in the MCAO model (Otero-Ortega et al, 2020, PMID: 32075692; Venkat et al, 2020, PMID: 32889008; Dumbrava et al, 2022, PMID: 34757568).

3. How did EVs affect the damaged brain—by reducing inflammation or providing nutritional support? What differentiates single-dose from multi-dose treatments? The authors should explain the underlying mechanisms supporting their findings.

We acknowledge that it is very important for the field to understand the underlying cellular and molecular mechanisms that improve recovery with EVs. We have now added this discussion point to our manuscript (lines 595-599). 

4. The authors should specify the number of animals used in behavioral tests, given the substantial standard deviation observed in Figs 3 and 4.

We thank the reviewer for the comment. As suggested, in addition to Table S1 (see below), the number of rats used for the behavioral tests has been added to the Methods section (lines 136-138) and Fig legends (Figs 2–5 and S1–S5).

Minor Revisions:

1. In Fig 1b, molecular weight ladders are not included in the Western blot (WB) images.

We thank the reviewer for the comment. Molecular weight ladders have been included in Fig 1b. 

Reviewer #2

This manuscript carefully investigates the effects of handling and/or intranasal EVs on the recovery after MCAO in rats, providing solid evidence that intensive behavior testing alone improves recovery after stroke, which could be enhanced after intranasal single dose of EVs. The authors have done rigorous experiments addressing this, while some concerns are needed to be addressed to fully clarify the statement.

Major points:

1. Fig. 2: effect of handling on neurological function after MCAO

The authors have shown that intensive behavior tests improve scores in mNSS as 28 dps. This is quite meaningful to the field since behavior tests have been typically used to assess performance after stroke and the contribution to stroke recovery has been overlooked. To further address the temporal effect of handling, it would be interesting to conduct these experiments at different timepoints, for example D7, D14, D21 dps like Fig. 3-4.

As sensorimotor tests are frequently used in preclinical stroke studies, and rehab is frequently given to humans after a stroke, we only focused on the handling component of the paper due to extensive behavior instead of pure spontaneous recovery without handling. We agree that it would be interesting to know what happens when only mNSS is measured at 7, or 14, or 21 dps in non-handled rats. In relation to this question, a previous publication mentioned in our manuscript has shown spontaneous recovery after MCAO in rats when only exposed to the neurological score test (Heras-Romero et al., 2022, PMID: 34563674). In this study, neurological deficits decreased by 7 dps, progressively improving by 14 dps, and plateauing by 21 dps. 

2. Fig. 3-4: effect of EVs (single and multiple doses) on neurological function after MCAO

The authors further explored whether intranasal EVs could enhance the handling-mediated neurological improvement after stroke at different time points, and found that single dose, instead of multiple doses showed synergistic effect, which might result from the lack of efficacy treated with multidose EVs according to the authors’ discussion. It would be important to address some concerns to further support the current statement.

2.1. EVs presence in the brain? The authors have nicely observed the presence of EVs 6 hours after a single dose (2.4 x 10^9 vesicles). How about the time course? It would be interesting to see if these EVs persists in the brain since most improvement after a single dose were observed in D28 dps (Fig. 4a-c). In addition, does EVs present in brains from MCAO rats after the lower multiple doses (0.8 x 10^9 vesicles)? A lower presence of EVs may explain the undetectable effects in stroke (Fig. 3).

Our studies demonstrated that EVs administrated intranasally can reach the brain and the injury site. Tracking the time course distribution of EVs in vivo presented some important caveats. Using fluorescent dyes to label EVs (as we did) is a great challenge for the field due to the characteristics of many commercial fluorescent dyes. Dyes can induce aggregation of EVs, or the dye's half-life cannot be long enough to be detected after a certain time (Yi et al, 2020, PMID: 31963931). Additionally, other cells can phagocytose labeled EVs, or EVs can release the dye and then be taken by other cells, keeping the dye long-term and producing misleading results (Yi et al., 2020, PMID: 31963931; Aimaletdinov et al., 2022, PMID: 36232613). We did not perform a long-term study using labeled EVs for these reasons. Nevertheless, previous studies using gold nanoparticles to label the EVs and computed tomography (CT) (Betzer et al, 2017, PMID: 28960957; Perets et al, 2019, PMID: 30761901) have demonstrated that intranasally administrated EVs are retained for at least 96h in the brain of a mouse model of focal stroke (ETH1-1 injection). These gold-labeled EVs were found around the ischemic region, as we saw in our study. Interestingly, they also saw that healthy control brains cleared gold-labeled EVs 24h post-administration. 

As the reviewer suggested, and as we discussed in lines 579-583, we speculate that the lack of efficiency of the multidose may be due to the low concentration of EVs per dose. Unfortunately, it is very difficult to assess if an accumulation of labeled EVs occurs in the peri-infarct area at 28 dps using the multidose for the characteristic of the dye explained above. 

2.2. Comparison between single dose and multiple doses of EVs on neurological function after MCAO? The authors have done separate experiments to compare the effect of single (Fig. 4) or multiple doses (Fig. 3) of EVs to handling alone on neurological function. Considering the baseline of most behavior tests may vary, it would be interesting to see how the dose of EVs may affect the handling-mediated neurological function in one set of experiment, or at least discusse in an integrated data from Fig. 3-4.

After integrating the data from Figs 3 and 4, comparing the “Handled + EVs multi” versus the “Handled + EVs single” (new Fig S5), no differences were found between these two groups in the mNSS at 28 dps (Figs S5a,b), percentage of ipsilateral hemisphere damage (Fig S5c), total ischemic volume (Fig S5d), and edema (Fig S5e) at 28 dps. This corroborates our modest improvement with the single dose (Fig 4a). It also emphasizes that the main recovery is largely due to handling. These findings have been added to the Results section (line 489) and Discussion (line 575).

2.3. Statistics analysis of EVs on neurological function after MCAO? The authors have conducted 2-way ANOVA to include both time and treatment (veh or EVs) to assess if there is a significant improvement, and nicely provided the raw data for the comparison in all individual days in the supplement files. However, it would be necessary to include the P value for a row and column factor (time and treatment here) in a 2-way ANOVA test. This would be more relevant to suggest if an overall improvement exist over time and/or after EVs treatment, in addition to the improvement in specific days.

We thank the reviewer for bringing up this important point. Results for each ANOVA test have been added to Table S5.

3. MRI analysis on the volume of the injury or edema

3.1. The authors have shown that either handling or EVs treatment had no effect on changes in injury volume after MCAO in rats, in contrast to the neurological function (Fig. 2-5). In Fig. 2c-d, less ipsilateral hemisphere damages were observed at D28 dps, while the representative images didn’t show a less bright volume. It would be necessary to clarify how the damage area were calculated in the Methods, and make sure to include representative images. Similar in Fig. 3-5.

See response to Reviewer #1 Point 1. For further clarification, we did not quantify the damage by measuring the brightness but by calculating the total volume after profiling the contour of the ischemic core in each coronal section. 

Minor points:

1. Cohorts and experimental groups in Methods:

The authors described that three different cohorts were established (line 133-135). In fact, the authors have done 5 groups rigorously to evaluate the effects of both handling and EVs treatment on stroke recovery (Table S1). It would be nice to also include the group information in Methods to make it easier to follow.

Number of rats used for the behavioral tests has been added to the Methods section (lines 136-138) and Fig legends (Figs 2–4 and S1–S4).

Reviewer #3

This is a well written manuscript, I feel comfortable reading it. However, there are some major concerns about the Figs and the statistical analysis.

1. Fig 1B. The western blot should include the markers showing the MW of the band.

We thank the reviewer for the comment. Molecular weight ladders have been included in Fig 1b.

2. Fig 1E. The Hoechest staining seems to be out of focus in the contralateral image, the authors needs a better image and do the quantification of the percentage of ExoGlow+ cells.

We have now included quantifications of the average size of ExoGlow+ particles in both hemispheres. Methods (lines 360-371, 374), Results (line 393), and Fig 1f have been updated. New fluorescence images have been added to Fig 1e.

3. Fig 2D. In the test, the authors wrote "No differences were found between handled and non-handled rats in the percentage of ipsilateral hemisphere damage." The point is "no difference". Therefore, the authors should compare "handled" vs "non-handled", not "2dps" vs "28dps" in quantification.

We apologize if this was not clear. All the raw data for the post-hoc tests for time and treatment are included in Table S5. We have also included the p-value for the row (time) and column (treatment) factors in Table S5. Comparison between handled and non-handled is mentioned in lines 422-424, and shown in Fig 2 and Table S5.

4. Fig 2D. The authors should discuss why handling improve neurological function, but not the ipsilateral hemisphere damage. What are the link between these two tests and what could be the potential explanation that two assays shows different results.

Traditionally, a general idea in the stroke field has been that a reduction of ischemic damage was needed to obtain sensorimotor recovery. Most of these results and conclusions were based on macroscopy-histologic examination of the tissue, using techniques such as TTC (2,3,5-triphenyl tetrazolium chloride) or H&E (hematoxylin and eosin) staining, which do not allow in vivo longitudinal examinations of the ischemic core from the same animal. In our experience, and due to the intrinsic variability of each MCAO stroke between animals, an MRI allows a better follow-up of the injury progression in a single animal. It allows better inclusion criteria, where only animals with similar strokes are included in the study. 

To add clarity about these results and conclusions, we have added more details to the previous discussion (lines 636-652).

5. Fig 3. The authors are comparing handling vs handling+multi EVs in this Fig, but in every panel the authors labels significant difference between "2dps and 28dps", but not "handling vs handling+multi EVs" this is quite misleading to readers.

For transparency, we included all the raw data from our statistical analyses (t-test, 2-way ANOVA and post-hoc tests) in Table S5. We performed all the statistical analyses to see if animals recovered over time and/or after EVs treatment. For clarity, our graphs did not show “non-significant” results for all post-hoc comparisons (time and treatment). Additionally, we only showed changes between the beginning of the treatment (2 dps) and the endpoint (28 dps) in the graphs to simplify the representation of data. To note, we performed other statistics to check possible improvement on specific days, and these results are shown in Table S5. 

***

We hope that with these revisions, the manuscript will now be found acceptable for publication in PLOS One.

Please do not hesitate to contact us if we can further assist you.

Sincerely,

Elena Blanco-Suarez, PhD 

Lorraine Iacovitti, PhD

---

## [Decision Letter · Decision Letter 1]

6 Sep 2024

PONE-D-24-14380R1Recovery after human bone marrow mesenchymal stem cells (hBM-MSCs)-derived extracellular vesicles (EVs) treatment in post-MCAO rats requires handling associated with repeated behavioral testing.PLOS ONE

Dear Dr. Blanco-Suarez,

Thank you for submitting your manuscript to PLOS ONE. After careful consideration, we feel that it has merit but does not fully meet PLOS ONE’s publication criteria as it currently stands. Therefore, we invite you to submit a revised version of the manuscript that addresses the points raised during the review process.

We look forward to receiving your revised manuscript.

Kind regards,

Hansen Chen

Academic Editor

PLOS ONE

**Journal Requirements:**

**Additional Editor Comments:**

Dear Authors,

Please address the reviewers' comments for the minor revisions. If you prefer the current title, you are not required to change it. Thank you!

Reviewers' comments:

Reviewer's Responses to Questions

**Comments to the Author**

1. If the authors have adequately addressed your comments raised in a previous round of review and you feel that this manuscript is now acceptable for publication, you may indicate that here to bypass the “Comments to the Author” section, enter your conflict of interest statement in the “Confidential to Editor” section, and submit your "Accept" recommendation.

Reviewer #1: All comments have been addressed

Reviewer #2: All comments have been addressed

Reviewer #3: (No Response)

2. Is the manuscript technically sound, and do the data support the conclusions?

Reviewer #1: Yes

Reviewer #2: Yes

Reviewer #3: Partly

3. Has the statistical analysis been performed appropriately and rigorously? 

Reviewer #1: Yes

Reviewer #2: Yes

Reviewer #3: Yes

4. Have the authors made all data underlying the findings in their manuscript fully available?

Reviewer #1: Yes

Reviewer #2: Yes

Reviewer #3: Yes

5. Is the manuscript presented in an intelligible fashion and written in standard English?

Reviewer #1: Yes

Reviewer #2: Yes

Reviewer #3: Yes

6. Review Comments to the Author

**Reviewer #1: **The authors developed a treatment based on hBM-MSCs-derived extracellular vesicles to reduce the damage size after MCAO. The revised manuscript is well done. The data fully support their view. I recommend removing "behavioral testing" from the title.

**Reviewer #2: **All major concerns have been addressed.

Minor points:

1. mNSS at D7, D14, D21 dps in handled rats

The authors have shown that intensive behavior tests improve scores in mNSS as 28 dps. To further address the temporal effect of handling, it would be interesting to measure mNSS at different timepoints, for example D7, D14, D21 dps. Alternatively, please include previous studies in Discussion.

2. Please discuss the presence of EVs in the brain

The author did not perform a long-term study using labeled EVs for technical limitations, and provided previous studies to support the likely presence of EVs in the brain (Betzer et al,

2017, PMID: 28960957; Perets et al, 2019, PMID: 30761901). It would be necessary to include this presumption in Discussion.

**Reviewer #3:** The abstract is too long in the current version. The authors described too much detail in the abstract. In the last sentence of the manuscript: These results show the importance of rehabilitation in combination with other treatments and highlight how intensive behavioral testing might influence functional recovery after stroke, especially when other treatments are also given. Here, "other treatments" seems to general, this kind of sentence could be present in discussion, but is quite in appropriate in abstract.

The authors only showed handling+single EV can improve mNSS and Beam Balance Test, the difference is mild. Therefore, the effect of EV is not solid, and cannot be concluded scientifically.

7. PLOS authors have the option to publish the peer review history of their article (what does this mean?). If published, this will include your full peer review and any attached files.

Reviewer #1: No

Reviewer #2: No

Reviewer #3: No

---

## [Author Response · Author response to Decision Letter 1]

26 Sep 2024

Reviewer #1 

The authors developed a treatment based on hBM-MSCs-derived extracellular vesicles to reduce the damage size after MCAO. The revised manuscript is well done. The data fully support their view. I recommend removing "behavioral testing" from the title.

Response: As suggested by the Reviewer, “behavioral testing” has been removed from the title. 

Reviewer #2

Minor points:

1. mNSS at D7, D14, D21 dps in handled rats

The authors have shown that intensive behavior tests improve scores in mNSS as 28 dps. To further address the temporal effect of handling, it would be interesting to measure mNSS at different timepoints, for example D7, D14, D21 dps. Alternatively, please include previous studies in Discussion.

Response: We thank the Reviewer for the comment. In our study, we observed that handled rats started their recovery in the mNSS at 7 dps, plateauing around 14 or 21 dps depending on the group (Fig 3 and Fig 4, Table S5). These results have been added to the manuscript for clarity (lines 446, 477, 557 in unmarked manuscript). As suggested, we added previous studies in our Discussion (lines 557-561 in unmarked manuscript) to explain how the lack of handling affects the mNSS at 7, 14, or 21 dps.

2. Please discuss the presence of EVs in the brain

The author did not perform a long-term study using labeled EVs for technical limitations, and provided previous studies to support the likely presence of EVs in the brain (Betzer et al, 2017, PMID: 28960957; Perets et al, 2019, PMID: 30761901). It would be necessary to include this presumption in Discussion.

Response: As suggested by the Reviewer, we added this part to the Discussion (lines 617-626 in unmarked manuscript).

Reviewer #3

The abstract is too long in the current version. The authors described too much detail in the abstract. In the last sentence of the manuscript: These results show the importance of rehabilitation in combination with other treatments and highlight how intensive behavioral testing might influence functional recovery after stroke, especially when other treatments are also given. Here, "other treatments" seems to general, this kind of sentence could be present in discussion, but is quite in appropriate in abstract.

Response: We appreciate the Reviewer's suggestions. The abstract has been made shorter and more clarity has been added to the conclusion in the abstract: “These results show the importance of rehabilitation in combination with other treatments such as EVs, and highlight how extensive behavioral testing might influence functional recovery after stroke” (line 47 in unmarked manuscript).

The authors only showed handling+single EV can improve mNSS and Beam Balance Test, the difference is mild. Therefore, the effect of EV is not solid, and cannot be concluded scientifically.

Response: We agree with the Reviewer’s comment. We have added more clarity throughout the manuscript to not overstate our results (lines: 478, 480, 484, 492, 546, 570, 575, 627 in unmarked manuscript).

We hope that with these minor revisions, the manuscript will now be found acceptable for publication in PLOS One.

Please do not hesitate to contact us if we can further assist you.

Sincerely,

Elena Blanco-Suarez, PhD 

Lorraine Iacovitti, PhD

---

## [Editor Report · Decision Letter 2]

4 Oct 2024

Recovery after human bone marrow mesenchymal stem cells (hBM-MSCs)-derived extracellular vesicles (EVs) treatment in post-MCAO rats requires repeated handling.

PONE-D-24-14380R2

Dear Dr. Blanco-Suarez,

We’re pleased to inform you that your manuscript has been judged scientifically suitable for publication and will be formally accepted for publication once it meets all outstanding technical requirements.

Kind regards,

Hansen Chen

Academic Editor

PLOS ONE
---

## [Editor Report · Acceptance letter]

9 Oct 2024

PONE-D-24-14380R2 

PLOS ONE

Dear Dr. Blanco-Suarez, 

I'm pleased to inform you that your manuscript has been deemed suitable for publication in PLOS ONE. Congratulations! Your manuscript is now being handed over to our production team.

Kind regards, 

on behalf of

Dr. Hansen Chen 

Academic Editor

PLOS ONE